# Prioritising Organisational Factors Impacting Cloud ERP Adoption and the Critical Issues Related to Security, Usability, and Vendors: A Systematic Literature Review

**DOI:** 10.3390/s21248391

**Published:** 2021-12-15

**Authors:** Sayeed Salih, Mosab Hamdan, Abdelzahir Abdelmaboud, Ahmed Abdelaziz, Samah Abdelsalam, Maha M. Althobaiti, Omar Cheikhrouhou, Habib Hamam, Faiz Alotaibi

**Affiliations:** 1Department of Information Technology, College of Computer and Information Sciences, King Saud University, Riyadh 11461, Saudi Arabia; 2School of Electrical Engineering, Faculty of Engineering, Universiti Teknologi Malaysia, Skudai 81310, Malaysia; 3Department of Information Systems, Faculty of Science and Arts, King Khalid University, Muhayel Aseer, Abha 61913, Saudi Arabia; aelnour@kku.edu.sa; 4Faculty of Computing and Informatics, Universiti Malaysia Sabah, Kota Kinabalu 88400, Malaysia; ahmedmazar1977@gmail.com; 5Department of Computer Science, University of Khartoum, Khartoum 11111, Sudan; sama7_279@yahoo.com; 6Department of Computer Science, College of Computing and Information Technology, Taif University, Taif 11099, Saudi Arabia; 7Computer Embedded System Laboratory CES-ENIS National School of Engineers of Sfax, University of Sfax, Sfax 3038, Tunisia; omar.cheikhrouhou@isetsf.rnu.tn; 8Higher Institute of Computer Science of Mahdia, University of Monastir, Monastir 5000, Tunisia; 9Faculty of Engineering, Université de Moncton, Moncton, NB E1A 3E9, Canada; 10Faculty of Computer Science and Information Technology, Universiti Putra Malaysia (UPM), Serdang 43400, Malaysia; faiz.eid@hotmail.com

**Keywords:** enterprise resource planning (ERP), critical success factor (CSF), cloud computing, cloud enterprise resource planning (CERP), adoption, security, usability, vendors, impacting factor, integrative model

## Abstract

Cloud ERP is a type of enterprise resource planning (ERP) system that runs on the vendor’s cloud platform instead of an on-premises network, enabling companies to connect through the Internet. The goal of this study was to rank and prioritise the factors driving cloud ERP adoption by organisations and to identify the critical issues in terms of security, usability, and vendors that impact adoption of cloud ERP systems. The assessment of critical success factors (CSFs) in on-premises ERP adoption and implementation has been well documented; however, no previous research has been carried out on CSFs in cloud ERP adoption. Therefore, the contribution of this research is to provide research and practice with the identification and analysis of 16 CSFs through a systematic literature review, where 73 publications on cloud ERP adoption were assessed from a range of different conferences and journals, using inclusion and exclusion criteria. Drawing from the literature, we found security, usability, and vendors were the top three most widely cited critical issues for the adoption of cloud-based ERP; hence, the second contribution of this study was an integrative model constructed with 12 drivers based on the security, usability, and vendor characteristics that may have greater influence as the top critical issues in the adoption of cloud ERP systems. We also identified critical gaps in current research, such as the inconclusiveness of findings related to security critical issues, usability critical issues, and vendor critical issues, by highlighting the most important drivers influencing those issues in cloud ERP adoption and the lack of discussion on the nature of the criticality of those CSFs. This research will aid in the development of new strategies or the revision of existing strategies and polices aimed at effectively integrating cloud ERP into cloud computing infrastructure. It will also allow cloud ERP suppliers to determine organisations’ and business owners’ expectations and implement appropriate tactics. A better understanding of the CSFs will narrow the field of failure and assist practitioners and managers in increasing their chances of success.

## 1. Introduction

Cloud computing is becoming more widely accessible, and businesses are increasingly trusting providers, allowing them to keep critical corporate information in the cloud. This, along with the manufacturers’ belief that the cloud is the best place to accommodate all long-term computer systems, is why cloud-based ERP has grown in popularity in recent years. According to studies on the ERP system industry [1,2], ERP systems that are cloud-based or hosted in the cloud increased their market share from 23 to 51% between 2015 and 2019. This suggests that the migration from on-premises to cloud-based ERP is moving rapidly. Over the previous year, according to a report released in 2016 [3], 44% of ERP systems deployed as cloud-based solutions. Cloud-based ERP can be set up in different ways and with varying levels of responsibility [4]. The first option is a software-as-a-service (SaaS) model. This means that the cloud vendor will be in charge of almost everything, including the database, infrastructure of the network, hardware, and applications. The only thing the firm needs to do is to configure the system and pay for the usage [5]. SaaS solutions are commonly hosted in a multi-tenant cloud system, where several distinct solutions can be implemented and hosted using the same hardware and infrastructure. Thus, every implementation will be unique within the system rather than in various virtual segregations [6].

The first goal of the study’s research was to prioritise the various variables driving cloud ERP adoption in organisations, to create better knowledge of cloud ERP clients in this market. The variables driving cloud ERP adoption differ by industry, depending on the size of the firm and the type of industry [7]. As a result, organisations need to identify and prioritise the variables driving the adoption of cloud ERP. The second objective of this research was to identify the critical issues in the adoption of CERP from a security, usability, and vendor perspective.

The research questions addressed in this study are:**a.** **What are the key success factors for the adoption of cloud ERP systems?****b.** **What are the key challenges for the adoption of cloud ERP systems in terms of security, usability, and vendors?**

According to the above research questions, the objectives of this study are as follows:To prioritise the innovative factors that will promote organisations’ adoption of cloud ERP;To identify the security factors that will either impede or promote organisations’ adoption of cloud-based ERP;To identify the usability factors that will either impede or promote organisations’ adoption of cloud-based ERP;To identify the vendor factors that will either impede or promote organisations’ adoption of cloud-based ERP.

The study’s importance and contributions are as follows. Firstly, the study provides an overview of cloud ERP systems in organisations, contributing to understanding of the benefits, success factors, and main drivers of ERP cloud adoption. Secondly, the existing legal issues of cloud ERP systems are analysed in terms of security issues, usability issues, and vendor-related issues. Next, this research used the SLR method to synthesise the available evidence, identify gaps in the current research, and provide a framework for directing future research efforts. Finally, this study presents the critical issues in the adoption of CERP in an understandable model. Accordingly, this study acts as one of the few studies to assess the effects of the crucial CSFs for ERP adoption success, as well as to identify the primary critical issues. As a result, the understanding of the influential factors and the critical issues in the adoption of cloud ERP systems will increase, to the benefit of organisations.

This study is structured in eight sections. Section 2 describes the background details of ERP systems, cloud computing, and the issues related to security, usability, and vendors in the adoption of cloud ERP systems. Section 3 explains the research methodology utilised in this study. Section 4 provides an evaluation of the literature, conducted on the basis of search terms and databases. Section 5 provides the results of the factors impacting adoption and critical issues related to security, usability, and vendors of cloud-based ERP systems, acquired using the SLR approach, followed by a discussion of the research model. The implications of the study are presented in Section 6. The contributions and research gaps are described in Section 7 of the study. Future research directions are finally provided in Section 8.

## 2. Theoretical Background

In this section, we provide a theoretical foundation for both of this paper’s primary themes, namely prioritising success factors for adoption of cloud enterprise systems and the main challenges for cloud ERP systems in terms of security, usability, and vendors. We describe the cloud ERP system, the benefits of cloud computing services, and special issues with respect to security, usability, and the vendors of CERP systems.

### 2.1. Cloud Enterprise Resource Planning (ERP) System

Cloud ERP is a type of enterprise resource planning (ERP) system which works on a vendor’s cloud platform rather than an on-premises network, enabling businesses to obtain data over the Internet [8]. ERP systems combine critical financial and operational company activities, offering a centralised data source for the organisation’s various departments, as well as support for sourcing, production, distribution, and fulfilment. Given its importance, any ERP system must be easily available to all business sectors, regardless of location, and must give a consistent, up-to-date view of the data. These needs are met by cloud-based ERP as a service [9]. ERP software that is implemented on the cloud is known as “cloud ERP software”. Virtualisation and load balancing are used in the majority of (if not all) cloud technologies. Settings are utilised to spread applications over several servers and database resources. Cloud ERP is positioned as a game-changing method of deploying ERP applications. It offers scalable, flexible, cheap, adaptive, and efficient solutions [10]. Cloud ERP has supplied essential company data with great success as corporate management software [11].

### 2.2. The Benefits of Cloud ERP Systems for Organisations

There are several advantages to using cloud ERP systems, but not all cloud ERPs are the same. Organisations should understand the many types of cloud ERP models in order to identify which model best aligns with their strategy, workloads, and security requirements, thus adopting the best cloud ERP solution. The following are some benefits of cloud ERP systems for organisations [8]:Avoiding the need for paying for all computer platforms, such as hardware platforms and data server platforms;Cutting IT services and support, since the data centre offers IT support;Replacing paying ahead for application software licensing with a monthly charge;Minimising the expense of support and maintenance, since these are provided by the vendors;Paying a fixed monthly charge that allows businesses to spend their funds on other business activities;Speeding up the use of systems, since no hardware or software needs to be installed on either servers or consumer devices;Avoiding attacks on the company’s server since the data are kept in the cloud rather than locally.

### 2.3. Growth of CERP Adoption

Cloud technology adoption is increasing globally as businesses migrate away from on-premises technologies for delivering business efficiencies, on-demand services, network elasticity, and expanded network access. Statistics on the growth of cloud-based ERP solutions show a significant increase in cloud applications in the ERP market. Although ERP implementation can be difficult, data and statistics show that many businesses meet, if not exceed, implementation efficiency expectations. A report in 2020 produced by a market research engine provided an overview of the global manufacturing companies that have adopted cloud-based ERP and related technologies, including Oracle, IBM Corp, Microsoft Corp, SAP, Infor, Sage, Netsuite Inc, TOTVS, Unit4, and Syspro. Figure 1 below shows the statistics on the growth of CERP in different regions [12].

### 2.4. Security Issues of Cloud ERP Systems

Consistently with the available literature [11,13,14], this study is also concerned with security, which is defined as the degree to which a cloud ERP system is viewed as insecure for data storage, data exchange, and other business operations. In cloud computing applications (CCAs), security is concerned with more than simply responsibility, authorisation, and authenticity; it is also focused on issues such as business operations, catastrophe recovery, and information security [15]. The service providers must provide enough data availability and an adequate number of supporting employees for maintenance or for addressing difficulties, as requested by the service seekers. Proper precautions and agreements on service levels are the driving forces behind achieving specified availability levels [16]. The service provider’s promise of security is a significant motivator in CCAs. Furthermore, privacy protection and confidentiality problems are also covered by security, since all information and data are available to the service providers and might be utilised for illegal purposes, either inadvertently or on purpose [17]. Security directly adds to the dependability of cloud-based systems, and a dependable ERP system can provide solid security. As a result, it is necessary to create very effective secure cloud-based systems in order to increase the rate of cloud computing acceptance [10].

### 2.5. Usability Issues of Cloud ERP Systems

When usability factors were discussed in the literature, authors largely agreed to regard the usability of cloud enterprise systems as the total of efficiency, effectiveness, learnability, and user pleasure [18]. According to several research papers, software projects should spend at least 12% of their money on usability in order to enhance their efficacy by 100% [18,19]. The usability is a crucial element that influences software success; it has been evaluated as the most effective factor in the human–computer-interaction (HCI) knowledge field [20], and ISO/IEC 9126 defines it as an essential quality characteristic [21]. With any new technology or solution, especially cloud ERP solutions, there will always be a degree of difficulty and there is always something fresh to learn. The cloud is simple to understand; nevertheless, it will take time for IT teams to learn how to traverse the complexity of a hybrid architecture in order to get the most out of the system’s capability. [22]. Compared to traditional ERP systems, most organisations feel that using a cloud-based ERP saves cost and time, improves communication, and results in greater client cooperation in new business applications. According to [23], usability has been highlighted as one of the essential components of information-system-based innovation adoption. More sophisticated corporate innovation procedures, however, are contentious. The greatest impediment to rapid technological adoption is complexity.

### 2.6. Vendor Issues of Cloud ERP System

Overall, the cloud-based ERP and cloud services markets are still in their early stages. As a result, the quality of ERP systems in the cloud supplied by various providers might vary greatly [24]. If a firm is dissatisfied with its existing vendor’s cloud services, it may move to a different service provider. However, according to [25], switching to a new cloud-based ERP supplier may be difficult, for a variety of reasons. Firstly, because of the cloud-based infrastructure’s complexity, transferring ERP data from one vendor to another may be both costly and time-consuming. Secondly, some ethical constraints delivered by the present cloud vendors may make it challenging for a user firm to extract and transfer their ERP data to the servers before or after an existing service contract with another cloud provider. Furthermore, the adopted ERP system is expected to restructure and modify company business processes and structures, power distributions, and organisational culture [26]. Altering existing ERP software necessitates modifications in several management processes and operations in other organisations. As a result of these possible concerns and difficulties, a user firm may be unable to change their cloud-based ERP vendor, even if the service is poor. A subsequent examination of the literature revealed that these challenges, usually referred to as the vendor lock-in scenario, are frequently encountered in the cloud context [27]. Vendor lock-in and changeover fees have emerged as major impediments to the retirement of CERP systems.

## 3. Research Method

Okoli [28] identified a systematic literature review in the field of information systems as “a methodical approach of finding, analysing, and synthesising the current corpus of finished and documented work created by researchers, academics, and practitioners, that is explicit, comprehensive, and reproducible”. Furthermore, a systematic literature review (SLR) [29] is a method for identifying, assessing, and interpreting all findings from relevant studies linked to cloud ERP system adoption, as well as the special issues in terms of security, usability, and vendors related to the adoption of CERP. The goal of an SLR is to adhere to a certain plan and an unambiguous review method throughout the planning stage that leads to execution. This goal aids in the construction of retrieved results while reducing researcher bias.

## 4. Conducted Literature Review

There are a number of relevant papers identified in databases that are connected to the topic of organisational factors impacting cloud ERP adoption, cloud ERP system deployment and acceptance, and challenges for cloud ERP systems in terms of security, usability, and vendors. We found a total of 73 research publications, with just 29 publications related to cloud ERP systems. As shown in Figure 1, the study may be split into two aspects based on the linked articles: the factors impacting cloud ERP adoption and the challenges of cloud ERP systems adoption.

### 4.1. Search Strategy and Database

The SLR of this study used the method of Kitchenham and Charters [30]. A search strategy is a predefined collection of key words that is used to search a database. The search strategy incorporates the essential principles of the search question in order to produce accurate results. The search strategy will also account for all possible related search terms, keywords, and phrases. The databases searched to extract findings for this study included:IEEE Explore (IEEE);Science Direct (SD);Scopus;Springer Link (SL);Google Scholar (GS);ACM Digital Library (ACM);Wiley Online Library (WOL).

The SLR methodology encompassed all empirical studies with respect to the CSFs of cloud ERP adoption in organisations. The study chose research articles that looked at CSFs for cloud ERP adoption. The search string utilised advanced research to confirm relevant articles. The keywords defined key success factors for cloud ERP systems and challenges related to security, usability, and vendors for cloud ERP systems in all the databases mentioned above. The following keywords were used to search all databases: factors and determinants, cloud ERP, CSF, Special issue, security issues of cloud ERP, usability issues of CERP, vendors issue of CERP, cloud ERP adoption, and other related words such as implementation, adoption, and diffusion. The program yielded 73 research publications in total, spanning the years 2011–2020 (see Figure 2).

### 4.2. Inclusion Criteria and Selection of the Studies

To filter the relevant papers, the following selection criteria for inclusion were used:Selecting only publications that focused on cloud ERP adoption in organisations;Selecting only papers focused on special issues and challenges for cloud ERP systems, such as security, usability, and vendors in the adoption of cloud enterprise systems;Selecting papers published between 2011 and 2020;Selecting papers published in the English language only;Selecting only papers that were published in peer-reviewed journals and conference proceedings.

### 4.3. Exclusion Criteria

Exclusion criteria are those characteristics that disqualify prospective subjects from inclusion in the study. The following were the exclusion criteria for this study:Papers should not be about anything other than the research issues;Article publishing date: any publication prior to 2011 was omitted;Papers from non-academic databases;Duplicate articles discovered in digital libraries;Studies that did not include CSF, security issues, usability issues, and vendor issues of ERP cloud adoption;Papers based on poor analysis, such as unpublished papers, editorials, and opinions;Redundant papers, discussion panels, instructional summaries, technical reports, article summaries, interviews, and poster sessions.

### 4.4. Inclusion Screening

We read the abstracts of the seventy-three (73) studies to determine their information content relevant to the research topic as showed on Figure 3. In addition, we independently assessed the articles in parallel. We then skimmed through the full-text articles to assess the studies’ quality and eligibility. Disagreements in the findings of the reviewers were discussed and resolved. Thirty-five studies were considered relevant (see Table 1), and the full-text articles were obtained for quality assessment.

### 4.5. Data Extraction

Following the study questions, data were extracted to identify the factors impacting cloud ERP adoption in organisations, as well as the challenges for cloud ERP systems. Finally, each shortlisted candidate’s text was evaluated for relevance using the inclusion and exclusion criteria. We also evaluated the reported research quality by examining the rigor of each publication’s technique description. Following this stage, a final corpus of 35 publications was chosen. The sequence of these stages is depicted in Figure 4.

Table 1 above shows the quantity of candidate papers in total generated by the database, following the application of the exclusion criteria. Table 2 shows the list of 18 papers and their year of publication based on organisational factors impacting the adoption of cloud ERP systems, followed by 17 research publications on the special issues and challenges influencing adoption of cloud ERP systems. We included all articles that were obtained from various databases; any papers that were recovered from multiple databases were evaluated only once.

## 5. Results

The literature reported on four important topics related to identifying the critical factors influencing cloud ERP adoption. Each topic was broken down into four sub-sections. Firstly, prioritisation of critical factors impacting the adoption of CERP systems based on their cited frequency with respect to success in CERP adoption is presented. Secondly, the distribution of critical issues in terms of their drivers is classified according to the categories of security, usability, and vendors. Then, Section 5.3 presents the validation of the research model. Section 5.4 finally outlines and discusses the critical issues related to security, usability, and vendors.

### 5.1. Prioritising Critical Factors Impacting the Adoption of Cloud ERP Systems

In the review, exclusion criteria were used to select 18 research papers to explore the factors impacting cloud ERP adoption. After all the studies had been included, we proceeded to extract the applicable information systematically and explicitly from each study. The frequency of the cited factors indicated how critical and significant they were for successful adoption. In response to research question 1, Table 3 presents the 16 critical factors in the adoption of cloud ERP systems and their priorities according to all the investigated research.

Table 3 above identifies sixteen impacting factors for cloud ERP adoption and their priority of influence. The literature study yielded a total of 16 factors impacting adoption of cloud-based ERP systems, including security of systems, senior management support, add-ons and customisation, ease of integration, and user education and training. The effectiveness of employees’ ICT skills was shown to be the most frequently reported influencing factor. The factors are discussed in more detail below.

*Security of the system* is the most widely cited impacting factor [13,24,34,35]. It is widely known that security breaches have serious repercussions for businesses, for example the loss of user account information, which allows attackers to access sensitive company information. Concerns about security while using cloud ERP include security breaches that jeopardise sensitive data, encryption, accountability, and maintenance difficulties [38]. As a result, security plans, policies, security tools, and processes play an important role in the successful adoption and implementation of CERP.

*Senior management support* is the second most frequently cited impacting factor affecting CERP system adoption [13,37,38]. The senior management is required to commit the necessary resources to a project and to accelerate the deployment of a new cloud ERP system. Support from the top management is a crucial success element for the business in the adoption of new technologies [61]. The senior management must support the adoption and explain the purpose and benefits of such systems to the firm and its people. Furthermore, a senior management with in-depth domain expertise may help companies expand their knowledge base [62], which can help organisations when they face problems during the adoption of CERP.

*Add-ons and customisation* represent the third most widely cited impacting factor [1,35,41]. To meet their automation needs, companies should look for cloud-based ERP suppliers with extensive customisation possibilities. Most organisations now recognise that modifying cloud-based ERP systems adds risk, time, and expense to the project. In reality, modifications, along with interfaces and data translation, are the primary technical risk areas in cloud ERP installations. According to [63], less than 20% of respondents in a recent poll built their ERP system with few or no modification, since customisations are always minor at first but gradually evolve into the technical problems that wreck these initiatives. Few cloud ERP installations have no adaptations but companies are very strict about justifying and managing even minor ones [64].

*Ease of integration* represents the next most widely cited impacting factor with regard to CERP systems [1,32,33]. In the case of cloud ERP solutions, ease of integration is critical. Businesses require a comprehensive CRM and ERP integration solution to optimise their company processes, since a lack of integration leads to an unproductive workplace. Integration guarantees that the logic of a cross-functional process is accurately represented. This means that data input into any of the functional modules (whatever module owns the data) are made available to any other module that requires it. This leads to considerable gains in terms of data consistency and integrity [65]. The automated data population (automatic data interchange among applications) that occurs between linked business components is the reason why ERP solutions are considered to be integrated.

*User education and training* is another widely cited impacting factor that is essential in the adoption of cloud-based services [35,39]. People are an important component of cloud ERP success. Organisations should ensure that on-the-job training is available, in which users apply real problems and examples in all of the ERP modules [66,67]. Change management includes training, and the most frequent technique is to “train the trainers”. Typically, software vendors or consultancy integrators teach the trainers, who are organisational employees [68]. This method is very beneficial because the business will then have skilled specialists in its workforce.

*Effectiveness of employees’ ICT skills* is also found to be critical for successful CERP adoption. Developments in Internet services and ICT technology have enabled ERP cloud suppliers to capitalise on new developing markets with novel packages of ICT solutions. These innovations are, in effect, driving businesses into new operational models. Moreover, as [69] found, 24% of 402 European enterprises said that having employees with the right skills to execute daily tasks on ERP cloud-based systems was their main obstacle. Moreover, the introduction of cloud computing strategies will introduce more uncertainty into the ICT labour market [70]. Cloud skills such as transmitting and receiving data using data-sharing programs and web storage systems such as OneDrive or Dropbox necessitate employees having a solid general technical understanding of cloud products and services [71].

### 5.2. The Challenges of Cloud ERP Adoption in Terms of Security, Usability, and Vendors

This section discusses the second research question of this study, which is to identify the critical issues for CERP adoption regarding security, usability, and vendors. Using the SLR approach, 12 critical issues were identified and associated with their main drivers of security, usability, and vendors of CERP. The section below discusses the special issues and challenges that were addressed by 17 research papers. Table 4 presents the distribution of special issues in cloud ERP adoption.

According to the SLR conducted on the special issues of cloud enterprise resource planning systems, the above table illustrates the most influential factors regarding CERP from the perspective of security, usability, and vendors. To identify possible variations, we chose to categorise the issues such that each publication had to be associated with at least one sub-concept in each of the three dimensions of security, usability, and vendors, based on the literature review findings.

### 5.3. Validation of the Research Model

To develop a comprehensive and scientifically valid model, according to Rahim [72], a four-step method was developed and implemented in this scenario to validate the research model. The first step was to remove natural duplication. This was the first stage in removing duplicates from the list of 12 drivers found in Table 4. Many of the drivers on the list are basically the same but are named differently. Step 2 was to identify empirically validated drivers. It is worth noting that empirical investigations do not support all of the drivers identified in the preceding stage; hence, this step is essential in developing an empirically supported model. Next, step 3 followed the empirical identification of supported drivers, while the final stage was to identify which drivers had a great deal of support. This stage was undertaken to find issues that were empirically validated in several research papers and were rated as strongly supported. As a result of this four-step methodology, an integrative model based on security, usability, and vendors consisting of 12 empirically validated factors impacting cloud ERP adoption was constructed, as described in the following section.

### 5.4. Discussion

According to a rigorous systematic mapping study, the above model (Figure 5) demonstrated a gap in the literature on cloud-based ERP adoption relative to security, usability, and vendor issues in the adoption of cloud-based enterprise systems. While relationships among critical issues and successful factors in the adoption of CERP were identified, in contrast, security, usability, and vendor issues were given less consideration in the current research on CERP adoption. This could be due to the factors associated with these issues, which are factors where people and organisations are more important than technical aspects. Furthermore, the criticality of these issues was not discussed from the viewpoint of security, usability, or vendors. Failure to properly address critical issues based on their importance may result in failure or in less-than-desired consequences [33]. As a result, the focus of this research is to fill this gap and investigate the importance of critical issues related to security, usability, and vendors in the adoption of CERP systems. However, the usability and vendor critical issues are unknown, as reflected in the small amount of frequently cited research. The section below discusses the main drivers for the research model in more detail.

#### 5.4.1. Security Issues

Surprisingly, security was found to be a critical success factor as well as a critical issue. This is due to the fact that the elements of data protection and security throughout the adoption and usage of cloud solutions are the most frequently mentioned CSFs. Ensuring cloud system security is a vital success element in CERP adoption and deployment. Data protection and information security are two critical elements that must be considered in the field of CERP adoption because security includes a variety of methods to protect illegal access and data damage. These measures may be organisational, related to personnel, or technical in nature. Precautionary measures to ensure information security are implemented by the cloud vendors and by the cloud users [45]. One of the major impediments to adopting cloud-based ERP is the data security risk, particularly the confidentiality and integrity of the organisation’s data. According to a recent IDC group study of 1100 businesses on the highest barriers for cloud-based ERP solutions, 60 percent of the organisations said confidentiality and security of data was their major worry when considering transferring their ERP systems to the cloud [73]. A previous study, for example, showed that one of the most important barriers to using cloud services for ERP systems for essential business applications is security [14,38]. The most prevalent issue in this evaluation is maintaining compliance with security, usability, and vendor requirements, and this issue is addressed in almost all the studied publications. Cloud security encompasses a wide range of problems, including privacy, confidentiality, and auditability [14]. Other researchers claim that cloud-computing security concerns are due to the possibility of third-party access to data, or difficulties with data transit and storage [34]. Hackers pose a massive risk to the security of data sent to and from the cloud; another key security risk confronted by businesses is identity theft [38]. However, this study found that the sub-drivers related to security were data leakage and loss, servers and the host, and data backup and recovery. These sub-drivers are in contrast to those in the study in [14], since their findings related to CERP were associated with three problems in information security: data confidentiality, encryption, and maintenance. Hacking activities can threaten data secrecy, and data encryption is critical since data loss can occur if the transmission is intercepted. Finally, IT infrastructure upkeep can avoid problems associated with hackers, viruses, and technological malfunctions. Concerns about security might arise for a variety of reasons. There are strategic reasons for protecting critical company information and regulations are growing ever more stringent [60].

#### 5.4.2. Vendor Issues

With respect to vendor drivers in cloud ERP systems, the legal concerns around data protection can be especially difficult for small businesses who lack structured legal teams. These concerns are mainly related to the CERP vendors since they are responsible for updates, server management, maintenance, and backups [50,52,54]. In other words, the advent of the cloud service model allows clients to have on-demand network access and share a collection of resources. Networks, servers, applications, and data storage are examples of such resources. It is difficult to determine the location of data storage in a cloud ERP solution or which laws apply to these data. The SaaS provider may be in Australia in a cloud supply chain, whereas the provider of the platform or infrastructure might be in Indonesia. In these cases, what are the regulations, and who is accountable if a disagreement arises? Close collaboration and assistance from an experienced vendor are key factors in effectively launching a cloud service, especially for novice businesses [45]. Future collaboration with cloud ERP vendors must centre on the selection process during the adoption phase [40]. Frequent meetings are held from the beginning to make certain that the systems built fit the criteria and are implemented in a low-risk setting [27]. Furthermore, confidence in the ability of the cloud ERP vendors to ensure continued functioning and support is seen as an issue and a necessary component of the partnership with cloud service providers.

#### 5.4.3. Usability Issues

Another source of concern discovered in the literature is connected to the usability of cloud ERP systems. Unsurprisingly, usability issues are top issues that might have a negative impact on the adoption of CERP [17,38,53]. According to the study in [74], 20 experts cited CERP system usability as a significant element in adopting ERP under a SaaS delivery model. Moreover, customers frequently made enquiries about the user interface, such as “is the user interface usable?” or “Is the UI the same as it is for on-premises solutions?” experts also stated that the client would like to “move to a more comfortable environment” and that the system should be “easy to use and intuitive”. According to experts, each time incorrect data is entered into a database, customers suffer. This finding confirms the findings of Almajali et al. and Nwankpa [75,76] that usability and user satisfaction improve overall organisational performance. It also concurs with many other academics that see usability as a crucial success component of the entire CERP project and the intention of businesses to utilise ERP software. When a company transitions to a cloud solution, usability issues with regard to cloud technology may face internal staff, which can cause obstacles such as security, customisation, resistance, and so on [67]. A lack of understanding is a significant challenge, and as a result, users face numerous difficulties throughout the adoption phase of cloud enterprise systems [19].

## 6. Implications of the Study

This research has important implications for practitioners, organisational managers, researchers, and CERP service suppliers with regard to formulating more effective policies and tactics to encourage adoption of cloud ERP systems. The established model is beneficial to service suppliers in understanding the critical issues in the adoption of cloud-based ERP, the interdependences among the multiple serious challenges, and their role in CERP deployment in the context of the organisation. Furthermore, companies need to understand the possible benefits of using cloud enterprise systems. In addition, CERP vendors should identify the variations in the offerings of cloud ERP services and focus on the concerns (such as add-ons and customisation, data backup and recovery, servers and the host, and time of implementation) of organisations, addressing these issues in order to maintain a positive connection with clients. Moreover, it is the responsibility of the CERP system vendors to guarantee that the organisational data are safe, secure, and available at all times. They must establish their dependability, building confidence, faith, and trust in their services. In addition, this research aids organisational managers, decision-makers, and policymakers in developing effective strategies and policies for deploying cloud enterprise system infrastructure and assessing the time period needed and the need for learning and training services if efficient adoption is to be considered.

## 7. Contributions and Research Gaps

There are weaknesses in innovation adoption of cloud ERP research regarding its CSFs. Moreover, there has been an excessive emphasis placed on cloud ERP adoption at the individual level and not enough on the security, usability, and vendor level. It is therefore evident that the theoretical foundation for this study was to consider prioritising the critical successful factors of cloud ERP adoption and the specific critical issues for organisations, such as security, usability, and vendor circumstances. It may be noted that there has not been much research conducted utilising SLR methodologies, especially in the domain of security, usability, and vendors in the adoption of cloud enterprise systems. For example, recent studies conducted by Tongsusai et al. and Huang et al. on CSFs were mainly on the implementation stages of cloud-based ERP systems. Hence, in this study, novel research into identifying and prioritising critical success factors for the adoption of cloud ERP in organisations and the related issues, especially from security, usability, and vendor perspectives, has been carried out. This it will aid in the development of new strategies or the revision of existing strategies and policies, in order to effectively integrate cloud ERP into cloud computing infrastructure.

## 8. Conclusions, Limitations, and Future Research

It is critical to identify and define the variables that have an impact on cloud ERP systems installation in order to motivate CERP adoption. The factors impacting cloud ERP systems adoption found in this investigation are listed in Table 3, sorted in order of their frequency in the literature. Sixteen variables were found from the literature as prioritised critical success factors in the adoption of cloud-based ERP. These are the most crucial elements for guaranteeing the effective adoption of cloud ERP solutions in organisations. In addition, this research mapped the security, usability, and vendor factors that influence CERP adoption, using an understandable model. The main limitation of this study is that while the gap regarding the CSFs was identified in the adoption stages of the ERP cloud-based systems, it could not be fully covered in the implementation or post-implementation stages due to the lack of publications, especially in the post-implementation phase. Thus, the next step in our research is to conduct an empirical study to analyse the CSFs, as well as the various organisational, environmental, technological, and personal aspects of the CERP systems at the different stages.

## Figures and Tables

**Figure 1 sensors-21-08391-f001:**
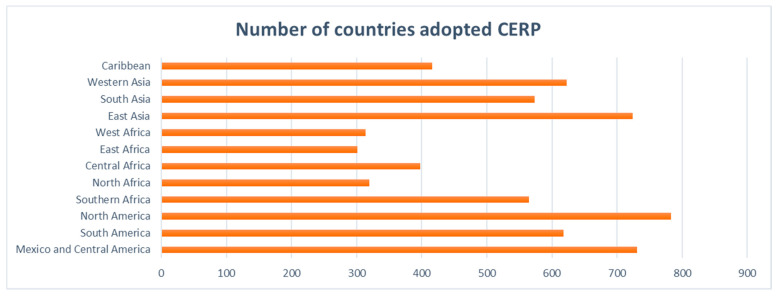
Statistics on the growth of CERP adoption in 2020: fastest-growing region—North America; slowest-growing region—East Africa.

**Figure 2 sensors-21-08391-f002:**
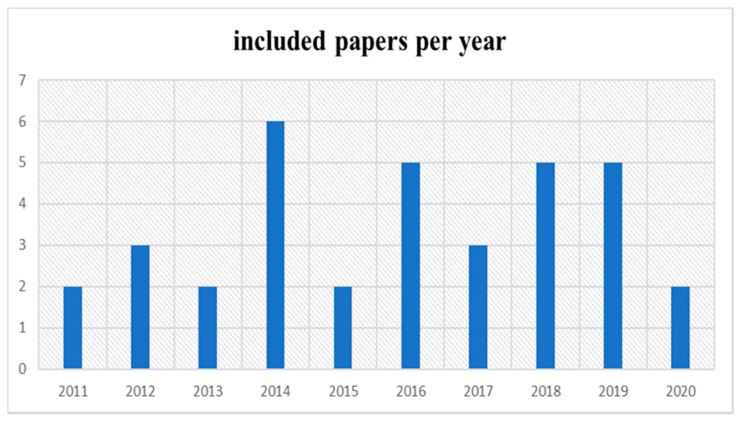
Distribution of related research according to year of publication.

**Figure 3 sensors-21-08391-f003:**
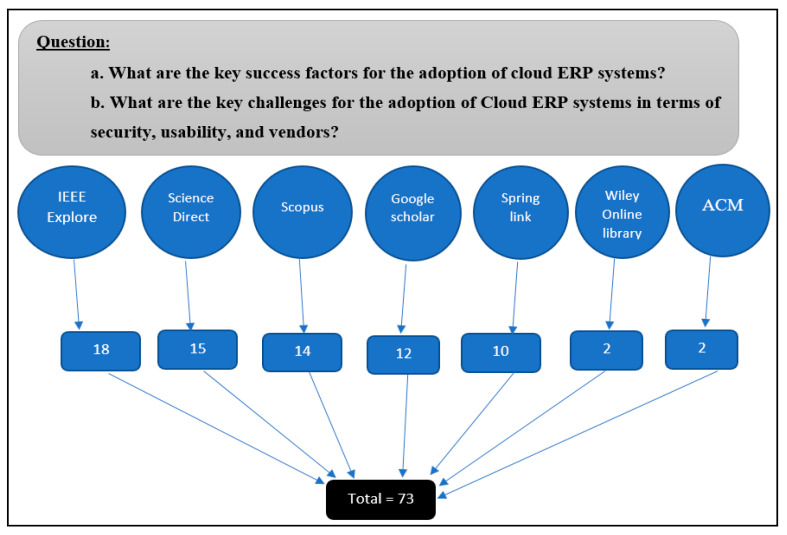
Search process and findings.

**Figure 4 sensors-21-08391-f004:**
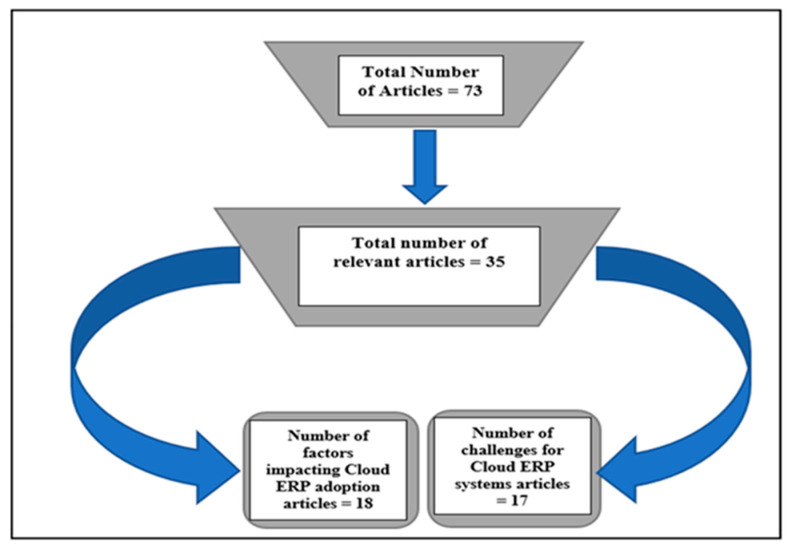
Diagram of systematic literature review.

**Figure 5 sensors-21-08391-f005:**
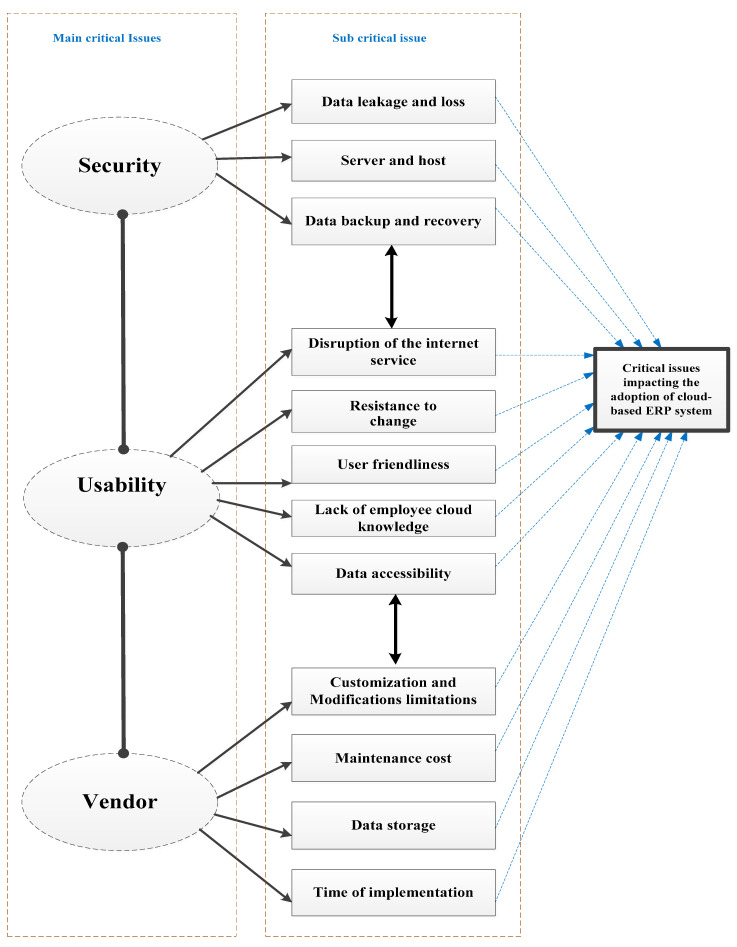
The integrative model for the critical issues in the adoption of cloud-based ERP systems from security, usability, and vendor perspectives.

**Table 1 sensors-21-08391-t001:** The total number of selected candidate papers.

IEEE Explore	Science Direct	Scopus	Google Scholar	Springer Link	ACM Digital Library	Wiley Online Library	Total
9	7	6	5	5	2	**1**	**35**

**Table 2 sensors-21-08391-t002:** List of papers identified from the literature and their year of publication based on organisational factors impacting adoption of CERP systems and the critical issues related to security, usability, and vendors.

**Papers Based on Organisational Factors Impacting Adoption of Cloud ERP Systems**			
**No.**	**Title**	**Year of** **Publication**	**Source**
1	Understanding the determinants of cloud computing adoption	2011	[31]
2	ERP on Cloud: Implementation strategies and challenges	2012	[32]
3	A framework for ERP systems in SME based on cloud computing technology	2013	[33]
4	Assessing the determinants of cloud computing adoption: An analysis of the manufacturing and services sectors	2014	[13]
5	ERP system adoption traditional ERP systems vs. cloud-based ERP systems	2014	[24]
6	Impacts on the organisational adoption of cloud computing: A reconceptualisation of influencing factors	2014	[34]
7	Factors influencing the adoption of cloud computing by small and medium enterprises in developing economies	2014	[35]
8	Implementation of ERP in cloud computing	2014	[36]
9	Determinants of cloud ERP adoption in Saudi Arabia: an empirical study	2015	[37]
10	Factors affecting the adoption of enterprise resource planning (ERP) on cloud among small and medium enterprises (SMES) in Penang, Malaysia	2016	[38]
11	Examining the critical success factors of cloud computing adoption in the SMEs by using ISM model	2017	[39]
12	Factors influencing cloud computing adoption by small and medium-sized enterprises (SMEs) In India	2017	[40]
13	Cloud ERP systems for small-and-medium enterprises: A case study in the food industry	2018	[41]
14	Factors affecting cloud ERP adoption in Saudi Arabia: An empirical study	2019	[42]
15	Determinants of ERP Systems as a Large-Scale Reuse Approach	2019	[1]
16	Role of cloud ERP and big data on firm performance: a dynamic capability view theory perspective	2019	[8]
17	Towards better understanding of determinants logistical factors in SMEs for cloud ERP adoption in developing economies	2019	[43]
18	Understanding potentials of cloud ERP adoption by large organisations: A Case Study	2020	[44]
**Papers Based on the Critical Issues Impacting the Adoption of Cloud ERP Systems Related to Security, Usability, and Vendors.**			
**No.**	**Title**	**Year of** **Publication**	
1	A selection framework for SaaS-based enterprise resource planning applications	2011	[45]
2	Challenges Involved in Implementation of ERP on Demand Solution: Cloud Computing	2012	[46]
3	In-house versus in-cloud ERP systems: a comparative study	2012	[47]
4	Factors affecting the adoption of cloud computing: an exploratory study	2013	[48]
5	Competition and challenge on adopting cloud ERP	2014	[49]
6	Technological, organisational and environmental factors influencing managers’ decision to adopt cloud computing in the UK	2015	[50]
7	Exploring the challenge impacted SMEs to adopt cloud ERP	2016	[51]
8	Benefits and challenges of cloud ERP systems–A systematic literature review	2016	[52]
9	An investigation of factors leading to the reluctance of SaaS ERP adoption in Namibian SMEs	2016	[53]
10	Moderating effect of compliance, network, and security on the critical success factors in the implementation of cloud ERP	2016	[14]
11	Identification of challenges and their ranking in the implementation of cloud ERP: A comparative study for SMEs and large organisations	2017	[54]
12	Factors influencing cloud-computing adoption in small and medium e-commerce enterprises in Jordan	2018	[55]
13	The determinant factors affecting cloud computing adoption by small and medium enterprises (SMEs) in Sabah, Malaysia	2018	[56]
14	Prioritising the factors affecting cloud ERP adoption–an analytic hierarchy process approach	2018	[57]
15	Software as a Service operation model in cloud-based ERP systems	2019	[58]
16	Cloud ERP in Malaysia: Benefits, challenges, and opportunities	2020	[59]
17	Cloud computing adoption and its impact on SMEs’ performance for cloud supported operations: A dual-stage analytical approach	2020	[60]
**Grand Total 35**			

**Table 3 sensors-21-08391-t003:** Prioritisation of the factors impacting the adoption of cloud enterprise resource planning.

Rank	CSF	Definition	Source
	**Security of the** **systems**	Refers to the assurance that the cloud-based ERP system provides a secure line of defence for the organisation against fraud and misuse, providing an unassailable network to workers anywhere, regardless of their location.	[13,24,34,35]
	**Senior management support**	Refers to the role of the senior management in an organisation when adopting cloud-based ERP in determining the resource allocation required and approving the project before execution.	[13,37,38,42]
	**Add-ons and** **customisation**	Refers to suppliers who provide integration with third-party add-ons or the opportunity to integrate particular modules for additional functionality in a cloud ERP system. A program’s customisation or setup is not inexpensive. As a result, organisations should look for a system that satisfies the majority, if not all, of their fundamental demands via out-of-the-box capabilities.	[1,35,41]
	**Ease of integration**	Refers to a cloud ERP service provider’s ability to seamlessly connect with other cloud-based ERP services based on the demands of the firms.	[1,32,33]
	**User education and training**	Refers to the level at which a company trains its staff before the implementation stage, in order to keep pace with changes and to improve or at least maintain the experience and capabilities of employees who use CERP systems.	[35,39]
	**Effectiveness of** **employees’ ICT skills**	Refers to the essential ICT skills that the employees should possess, especially cloud computing skills.	[39,42]
	**Service providers’ dependability**	Relates to the vendor’s reliability with regard to cloud-based ERP software applications.	[40]
	**Data backup and recovery**	Relates to the extent to which a cloud ERP service can swiftly return to operating in a safe manner following an unforeseen disruption.	[8]
	**Retention of data**	Determines if cloud ERP systems can preserve data when customers or cloud service providers alter or remove data. There might still be data left behind, potentially disclosing sensitive information to unauthorised parties.	[36]
	**Cost of software maintenance and upgrades**	Refers to the vendor’s costs for, and frequency of, upgrades and whether the charges for these changes are included in the original cost of the cloud ERP system.	[56]
	**Maintainability**	Refers to the capacity of cloud ERP service providers to make changes without interfering with the service or having a negative impact on the system.	[44]
	**Usability as** **perceived**	Refers to the degree to which consumers believe cloud ERP is simple to access, learn, and use.	[31]
	**Effectiveness of** **inventory and** **inventory carrying cost**	Refers to the delivering of one-time asset reduction (cost of the material stored), but also continuing reductions in inventory carrying costs, storage, handling, obsolescence, insurance, taxes, damage, and shrinkage. Cloud ERP systems enable clients to obtain information on costs, sales, and margins, allowing them to better manage their total material cost structure, leading to inventory savings of 20% or more.	[13]
	**Reliability of the** **Internet**	Relates to the dependability of an Internet connection and infrastructure, which encompasses connection to the Internet and access as well as Internet speed.	[42]
	**Government regulations and policies**	Refers to whether the government has a distinct policy on technological advances, in which case firms are more inclined to adopt them.	[43]
	**Use of latest IT** **technology**	ERP suppliers utilise the most recent advances in information technology. As a result, they quickly modify their systems to make use of cutting-edge technologies such as open-source software, client-server technologies, computer-assisted acquisition and logistics support, and e-commerce.	[35]

**Table 4 sensors-21-08391-t004:** Distribution and classification of critical issues in terms of their drivers.

Main Critical Issue	Related Sub-Issues	Definition	Source
Security	Data leakage and loss	Because CERP is interconnected, the data saved in the system may be used and shared by multiple organisational divisions. As a result, staff typically store several copies of critical company data on laptops or in flash memory. If any of these hardware items damaged, the threat of unauthorised access to the information stored on the device will increase. Furthermore, company staff may illegally steal data that are confidential and distribute them to others in order to increase their earnings.	[14,38,51]
	Server and host	When CERP data are managed by a third-party cloud service, the client firm has fewer controls over who has access to its critical data. Such a loss of control in a cloud environment inexorably leads to new data security risks in customer organisations. Whoever uses cloud ERP systems will have to put his/her company’s secrets on third-party servers, which may be exposed to spying, theft, or even hacking.	[60,61]
	Data backup and recovery	This is the factor that determines whether the cloud ERP vendor can maintain an overlapped copy of the data and how quickly the cloud ERP service can return to a healthy operating condition following an unanticipated disruption.	[57]
Usability	Disruption of the Internet service	When the Internet service is disrupted at the headquarters of the company that uses cloud systems, the system will stop working completely, becoming completely paralysed until the Internet service is restored.	[38,51,53,54]
	Resistance to change	Larger organisations with large IT and administrative resources may meet opposition from their main stakeholders. Administrators lose some control over procedures that become automated when ERP software is moved offsite. Furthermore, because the vendor is in charge, the IT department lose control over various operational activities, due to maintenance and infrastructure issues.	[48,60]
	User friendliness	Refers to user expectation of fast, friendly, and intuitive technology that is easy to learn and use.	[45,57]
	Lack of employee cloud knowledge	Cloud computing necessitates abilities that most IT staff members in established businesses lack. Cloud-based ERP technology enables more mobility and real-time collaboration, which eliminates the practice of delivering the incorrect version of a file. Furthermore, because the cloud is scalable, it is unlikely that businesses will run out of space.	[51,59,60]
	Data accessibility	Refers to the ability of employees to access cloud ERP systems from any place at any time via any device with Internet access, including a mobile phone, tablet, or laptop. This is an advantage to employees who frequently travel and need to access ERP software from a hotel or conference meeting room.	[57,58]
Vendors	Customisation and modification limitations	Companies that use the cloud systems will not be able to make their own modifications to the systems (customisation) freely because the systems are shared with other beneficiaries.	[46,50,52,54]
	Maintenance cost	ERP software is not cheap, and prices vary based on deployment method, number of users, and amount of customisation. Companies may also pay a monthly subscription charge to use the CERP system, where the system is hosted and maintained by the vendor on a third-party data server. This also means that maintenance cost is a factor that impacts the ERP (SaaS) pricing, implying monthly or yearly costs are paid per user.	[54,55,56]
	Data storage	The fact that all data will be kept in a data centre that is not near to the organisation may also be an issue. The CERP provider is likely to have a considerably larger security budget than other firms. This can be an issue with various data types since they are usually required by law to be stored inside the same nation as the organisation, and cloud vendors do not have data centres in all countries.	[49,57]
	Time of implementation	The time it takes to deploy large and complicated company software such as ERP systems varies from case to case. It might take anywhere between a few months and several years. The time required for implementation is determined by modifications and data translation, as well as by the number of desired modules, available resources, and deployment locations.	[47]

## Data Availability

Not applicable.

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
