# Peer review of "Prioritising Organisational Factors Impacting Cloud ERP Adoption and the Critical Issues Related to Security, Usability, and Vendors: A Systematic Literature Review"

_sensors, 2021, doi:10.3390/s21248391_

Round 1

Reviewer 1 Report

Prioritizing organizational factors impacting Cloud ERP adoption and the critical issues related to Security, Usability, and vendors: A systematic literature review

The goal of this study is to rank and prioritize the factors driving cloud ERP adoption on organization and to identify the critical issue in terms of security, usability, and vendors that impact adoption of cloud ERP systems. This research would also allow cloud ERP suppliers to determine organizations and business owner’s expectations and implement appropriate tactics. 

  1. What is you major debate point from the “prioritizing organizational factors impacting Cloud ERP adoption” ? Please underscore the scientific value added/contributions of your paper in your abstract and introduction and address your debate shortly in the abstract.
  2. The questions or objectives could choice one presented in your article. Abbreviations and acronyms are often defined the first time they are used within the abstract and again in the main text and then used throughout the remainder of the manuscript.
  3. I think these issue should proper address in the introduction “Security, Usability, and vendors”
  4. I would suggest the author to discuss these references in your context and references. Ming-Lang Tseng, Thi Phuong Thuy Tran, Hien Minh Ha, Tat-Dat Bui & Ming K. Lim(2021) Sustainable industrial and operation engineering trends and challenges Toward Industry 4.0: a data driven analysis, Journal of Industrial and Production Engineering, 38:8, 581-598, DOI: 1080/21681015.2021.1950227; and Ricardo Luhm Silva, Osiris Canciglieri Junior & Marcelo Rudek (2021) A road map for planning-deploying machine vision artifacts in the context of industry 4.0, Journal of Industrial and Production Engineering, DOI: 10.1080/21681015.2021.1965665
  5. How do you choose “Security Issues” “Usability Issues”… in your literature review section ? Why not others ?
  6. Basically, you should enhance your findings, limitations, underscore the scientific value added of your paper, and/or the applicability of your contributions/shortages and future study in this session.
  7. In general, I would appreciate to review this article. I am willing to see the revision version

Author Response

please see the attached manuscript 

Reviewer 2 Report

    This paper is interesting, current and creates multiple opportunities for further research in the future. The authors conducted a study that aimed to identify the factors (16 factors) that determine the adoption of cloud ERP within the organization and find critical issues (12 issues) related to security, utilization and vendors that influence the adoption of cloud ERP systems. It was based on a systematic analysis of the SLR literature that included 73 publications on the adoption of ERP in the cloud, from different sources. The results of the analysis lead to the construction of a model focused on three variables: security, usability and vendor’ characteristics that may have an influence on cloud ERP system adoption

Some suggestions in the process of revising this paper:

  • considering the large number of analyzed publications, I recommend the introduction of statistical data related to the increase in the number of companies from different countries that have chosen to adopt and implement Cloud ERP;
  • To be modified „2. theoretical Background ”

With minor modifications, I recommend publishing the article.

Author Response

Response to Reviewer 2 Comments#

Point 1: considering the large number of analyzed publications, I recommend the introduction of statistical data related to the increase in the number of companies from different countries that have chosen to adopt and implement Cloud ERP; To be modified „2. theoretical Background ”

Response 1: Thank you for this very important point of correction- section have been added in the (theoretical Background) to the revised manuscript – (See section 2.3)

Reviewer 3 Report

The paper deals with the current topic, the issue of ERP in recent years.

A slight issue is the non-embedding of ERP in the areas of Business, process management, or Industry 4.0. Here, I recommend expanding the introductory text with a paragraph such that the importance of ERP in the global context stands out. For example, the following resources can be used:

doi:10.3390/electronics10080959.

doi:10.3390/su12104303.

doi:10.3390/app9245405.

https://www.researchgate.net/publication/291836096_Level_of_process_management_implementation_in_SMEs_and_some_related_implications

The literature review contains all the essential literature related to the topic.

The authors work with all the major systems and databases of literature sources, but the text does not explain how they dealt with the overlap of these databases.

The thesis does not explain on what basis the Exclusion Criteria were defined.

Also, it is common in literature review methodology for authors to indicate the overall sieve of all articles and how they arrived at the desired article.

Overall, the article can be recommended for publication after incorporating the requirements.

Author Response

Response to Reviewer 3 Comments#

Point1: A slight issue is the non-embedding of ERP in the areas of Business, process management, or Industry 4.0. Here, I recommend expanding the introductory text with a paragraph such that the importance of ERP in the global context stands out. For example, the following resources can be used:

doi:10.3390/electronics10080959.

doi:10.3390/su12104303.

doi:10.3390/app9245405.

https://www.researchgate.net/publication/291836096_Level_of_process_management_implementation_in_SMEs_and_some_related_implication

Response 1: Thank you for this observation, suggested resources have been discussed and added to the reference list – (See reference number 2, 5, and 66 on manuscript).

Point 2: The literature review contains all the essential literature related to the topic.

The authors work with all the major systems and databases of literature sources, but the text does not explain how they dealt with the overlap of these databases. The thesis does not explain on what basis the Exclusion Criteria were defined. Also, it is common in literature review methodology for authors to indicate the overall sieve of all articles and how they arrived at the desired article.

Response 2: Thank you for this observation. Correction made accordingly and section have been added – (see section 4.3 and 4.4) and (Table 1 and 2) in the revised manuscript for the full-text article was obtained for quality assessment.

Round 2

Reviewer 1 Report

accepted